# Development of a Magnetic Compound Fluid Rubber Stability Sensor and a Novel Production Technique via Combination of Natural, Chloroprene and Silicone Rubbers

**DOI:** 10.3390/s19183901

**Published:** 2019-09-10

**Authors:** Kunio Shimada, Ryo Ikeda, Hiroshige Kikura, Hideharu Takahashi

**Affiliations:** 1Faculty of Symbiotic Systems Sciences, Fukushima University, 1 Kanayagawa, Fukushima 960-1296, Japan; 2Institute of Innovative Research, Tokyo Institute of Technology, 2-12-1 Ookayama, Meguro-ku, Tokyo 152-8550, Japan

**Keywords:** dimethylpolysiloxane (PDMS), polyvinyl alcohol (PVA), diene rubber, non-diene rubber, piezo effect, induced voltage, mechanical property, metal complex hydrate, silicone oil rubber, natural rubber, chloroprene rubber, sensor, adhesion, electrolytic polymerization, magnetic cluster, magnetic field, magnetic compound fluid (MCF), robot

## Abstract

Expanding on our previous report, we investigate the stability of a magnetic compound fluid (MCF) rubber sensor that was developed for a variety of engineering applications. To stabilize this sensor, we proposed a novel combination technique that facilitates the addition of dimethylpolysiloxane (PDMS) to natural rubber (NR)-latex or chloroprene rubber (CR)-latex using polyvinyl alcohol (PVA) by experimentally and theoretically investigating issues related to instability. This technique is one of several other novel combinations of diene and non-diene rubbers. Silicone oil or rubber with PDMS can be combined with NR-latex and CR-latex because of PVA’s emulsion polymerization behavior. In addition, owing to electrolytic polymerization based on the combination of PDMS and PVA, MCF rubber is highly porous and can be infiltrated in any liquid. Hence, the fabrication of novel intelligent rubbers using any intelligent fluid is feasible. By assembling infiltrated MCF rubber sheets and by conducting electrolytic polymerization of MCF rubber liquid with a hydrate using the adhesive technique as presented in a previous paper, it is possible to stabilize the MCF rubber sensor. This sensor is resistant to cold or hot water as well as γ-irradiation as shown in the previous report.

## 1. Introduction

Rubber sensors are valuable elements in a wide variety of engineering applications that affect many aspects of our daily life. The elasticity and flexibility of rubber have resulted in significant progress in several applications. To advance this field, we have investigated the mechanical, electrical and photovoltaic characteristics of a proposed magnetically responsive fluid called MCF involving rubber latex, the so-called magnetic compound fluid (MCF) rubber. This material exhibits hybrid properties during electrolytic polymerization under an applied magnetic field [1,2,3,4,5,6,7]. MCF rubber is a soft rubber that can be easily made using the latex of natural rubber (NR), isoprene rubber (IR), chloroprene rubber (CR), and butadiene rubber (BR) based on our proposed electrolytic polymerization procedure [1]. These rubbers contain C=C bonds so that they can be electrolyzed for vulcanization [1,4]. However, given that nitrile rubber (NBR) or styrene-butadiene rubber (SBR) potentially has a high viscosity, electrolytic polymerization is difficult. In addition, the MCF rubber can be used as a sensor by exploiting our proposed novel adhesion technique [8] whereby the MCF rubber is electrolytically polymerized in conjunction with a metal complex hydrate so that the metal electrodes strictly adhere to the MCF rubber. Based on the reaction of the MCF rubber with electromagnetic waves, a variety of engineering applications can be developed. In a previous report [9], the effects of γ-rays, infrared and microwaves on the MCF rubber sensor were evaluated, establishing the feasibility of using the MCF rubber sensor for sensing various solar and thermal sources, etc., related to energy harvesting. In particular, it is suitable for use as a sensor installed in a robot that operates in a nuclear reactor building.

However, one issue is the stability of the MCF rubber sensor. The stability of the changes of electrical properties with time is a significant problem. In the present report, we investigate the causes of the instability and deterioration in MCF rubber. Next, we resolve these problems by introducing a combination of non-diene rubbers. The non-diene rubbers include butyl rubber (IIR), ethylene-propylene rubber (EPM or EPDM), urethane rubber (UR), silicone rubber (Q), fluoro rubber (FKM), etc., which do not or seldom contain C=C bonds. Non-diene rubbers show superior weather, ozone, chemical, and oxidation resistance, less deterioration after years of usage under a variety of environmental conditions, and they can be used at any temperature. In addition, in terms of the sensitivity of a robot operated in a nuclear building, we also investigate in the present report the feasibility of an underwater MCF rubber sensor. To date, few studies have attempted to develop rubber-type water sensors in contrast with typical sensors including acoustic ones, etc. [10,11].

Regarding the combination of any material into a rubber, there have been many studies on the combination of fillers such as carbon, metal, etc. particles in NR [12], CR [13], NBR [14], SBR [15], Q [16,17] rubbers and in the combinations of NR and SBR [18,19,20,21], NR and BR [21], NR and EPDM [22]. However, in terms of the latter combination, vulcanization with sulfur has been used. As observed in the results for γ-irradiation effect on the MCF rubber sensor in the previous report, our electrolytic polymerization for rubber vulcanization without using sulfur results in some typical characteristics, for example, the enhancement of softness during the elongation. Therefore, the utilization of electrolytic polymerization in the combination of several rubbers is expected to be effective. In addition, given that non-diene rubber offers excellent resistance as previously mentioned, the combination of diene and non-diene rubbers is also suitable. However, fabrication of this combination is akin to mixing water and oil because each rubber is generally similar to water and oil, respectively. This means that the rubbers behave as hydrophilic and hydrophobic groups [23]. Therefore, we can apply the principle of emulsion polymerization to the combination as used in the mayonnaise production process. In the present report, we address and propose a novel rubber production process via the combination of NR and CR, and Q.

## 2. Stability

### 2.1. Factor That Influence Instability

Firstly, we will begin by discussing the factors that influence the instability of an MCF rubber sensor. In the previous study, we evaluated the secular stability of electrolytically polymerized MCF rubber sheets fabricated by sandwiching between opposing magnets [1]. However, it is typically necessary to measure the electric properties during sandwiching of the MCF rubber between the opposing electrodes. This electric property has secular stability. However, when the electrodes are separated, the stability deteriorates with time. Instability can be divided into two phases: (a) temporary instability during sensing that fluctuates with time; (b) instability over a long span of time leading to aging degradation. However, MCF rubber sheets must be considered because they are fundamental components of MCF rubber sensors. From this perspective, examples of factors that influence instability include: (1) the evaporation of water from the MCF rubber because water is substantially involved in the fabrication process; (2) the vulcanization of the MCF rubber occurs during the electrolytic polymerization process with surpassing this point, in contrast to electrolytic polymerization in our previous studies; (3) the deformation of the MCF rubber by the Mullins effect. Factors (1–3) are related to (a) and (b).

In the case of (1), water is involved and is trapped in the inner MCF rubber, even if electrolytic polarization is performed, because the NR and CR latexes contain water. Therefore, due to the evaporation of water from the electrolytically polymerized MCF rubber, the MCF rubber sensor can exhibit a secular change of its electrical and mechanical properties. This will be discussed in the following section.

In the case of (2), the MCF rubber liquid should be taken into account because it is fundamental in the production of MCF rubber sensors. In the previous study, we experimentally investigated the influence of the aggregation of magnetic particles and rubber molecules of MCF rubber liquid on their electrical characteristics when a magnetic field was applied [24]. The application of a magnetic field to a container immersed in the MCF rubber liquid resulted in the aggregation of particles and molecules to sediment. In the present report, we will introduce the result that the MCF rubber in the middle of vulcanization induces temporal instability by using an experimental apparatus with large electrodes gap as shown in Figure A1 in the Appendix A. The MCF rubber liquid consisted of 12 g Ni powder with particles on the order of microns and bumps on the surface (No. 123 by Yamaishi Co. Ltd., Noda, Japan), 3 g water-based MF with 50 wt% Fe_3_O_4_ (M-300, Sigma Hi-Chemical Co. Ltd., Tsutsujigasaki, Japan), 16 g NR-latex (Ulacol, Rejitex Co. Ltd., Atsugi, Japan) and 31 g of water. Using electrolytic polymerization, the MCF rubber is vulcanized as shown in Figure A2 in the Appendix A. The vulcanized MCF rubber grows from the anode surface towards the direction of the cathode as a crystal. The thickness of the vulcanized MCF rubber increases with time as shown in Figure A3 in the Appendix A. Isoprene molecules and magnetic clusters are aligned along the same direction as the electric and magnetic field lines, as explained in previous studies [1,4]. The structure can be considered to be approximately quasi-regular crystalline, but not a crystalline lattice such as in hexagonal close-packed, body-centered cubic, and face-centered cubic structures. The density of the aggregation at 0 mm was larger than that at 15 mm such that the thickness of the vulcanize MCF rubber at 0 mm was larger than that at 15 mm. As shown in Figure A4 in the Appendix A, as the thickness of the vulcanized MCF rubber increases, the electrical conductivity decreases. Therefore, if the vulcanization of the MCF rubber occurs during the electrolytic polymerization process without surpassing this point, the electrical signal from the MCF rubber sensor becomes unstable when the piezoresistivity of MCF rubber sensor is examined. This refers to usage whereby a generated electric current is measured via the application of a voltage with any electric source.

In the case of (3), we must consider the change of the electric current flowing in the MCF rubber based on its deformation. The change of the electric current of the MCF rubber by compression has been elucidated in previous studies [1,2,3]. In addition, this has been theoretically explained in other studies [5,24]. The relationship between the transmitted probability *T*, which represents the electric current flowing in the MCF rubber and the thickness *b*, is shown in Figure A5a in the Appendix A. The relationship between the dimensionless capacitance *C** and the thickness *b* is shown in Figure A5b in the Appendix A. *T* and *C** increase by enhancing *b*. *T* is considered to be re-expression of the electric current *I*, *C** applies to the capacitance *C*, and *b* applies to the deformation quantity. The motion of electrons described by the tunnel theory is categorized as electron transfer in the field of complex chemistry. The mechanism of the electron transfer is divided into two types based on the distance among the particles of Fe_3_O_4_ and Ni, and the molecules of polyisoprene: outer-sphere electron transfer reaction (OSETR) and inter-sphere electron transfer reaction (ISETR), as shown in a previous study [5]. Whether OSETR or ISETR is generated depends on the probability of the distance among the particles and molecules—OSETR takes place in the case of long distances and ISETR occurs for shorter distances. Decreasing *b* represents a contraction of the MCF rubber. As *b* becomes smaller, ISETR is dominant in the change of the electrical property of the MCF rubber due to the electric-chemical reaction. In contrast, as *b* gets larger, OSETR is dominant. As the distance among the particles and molecules is reduced by deformation of the MCF rubber, *I* and *C* increase as shown in Figure A5. The enhancement of *I* and *C* are presented as *ΔI* and *ΔC*, respectively. In the case of *ΔI* and *ΔC*, we can obtain the results as represented by Equations (1)–(3), together with Equation (A3) in the Appendix A. The notion is that *I* represents the amount of electron transfer and that *C* represents the amount of counter ion created by an anionic acceptor A^−^ and cationic donor D^+^. A and D are generated by particles and molecules with semiconducting roles by mixing and electrolytic polymerization as shown in Equations (A4)–(A7) in the Appendix A, which has been presented in a previous study [5]. However, the reaction occurs from the right-hand-side to the left-hand-side by irradiation of electromagnetic waves of γ-rays, microwave, light, etc. In the case of OSETR, which implies that the distance among the particles and molecules is large, the net reaction from the right-hand-side to the left-hand-side does not occur practically. In the case of ISETR, which implies that the distance among the particles and molecules is small, the net reaction from the right-hand-side to the left-hand-side occurs practically.

In the case of Equation (1), the voltage *V* increases with time. The amount of A^−^ and D^+^ is small because ISETR is dominant in the reaction as shown in Equations (A4)–(A7), due to the small distance among the particles and molecules. However, the amount of electron transfer is larger than that of the counter ion:(1)|ΔI|>|ΔC|

In the case of Equation (2), the voltage *V* with time is constant. The amount of electron transfer is the same as that of counter ion:(2)|ΔI|=|ΔC|

In the case of Equation (3), the voltage *V* with time decreases toward zero. The amount of A^−^ and D^+^ is large because of OSETR is dominant in the reaction as shown in Equations (A4)–(A7) due to the large distance among the particles and molecules. However, the amount of electron transfer is smaller than that of the counter ion:(3)|ΔI|<|ΔC|

Mullins effect is induced by the inner deformation among the particles and molecules of the MCF rubber. MCF rubber was determined to exhibit the Mullins effect in previous studies [1,24]. Therefore, the instability related to the aforementioned (a) and (b) occurs, such as the alternation of increasing and decreasing voltage and electric current of the MCF rubber according to Equations (1)–(3). As a result, the instability based on the deformation of the MCF rubber by Mullins effect also occurs.

Moreover, the instability of the MCF rubber is significant when utilized as a sensor. To prevent the aforementioned instability: (1) a possible idea is to deposit some material on the MCF rubber sensor. However, for the MCF rubber or MCF rubber sensor, this is useless because of the attempt to prevent the aforementioned instability (2) as follows. At first, we can clarify that by comparing the electrical property between the covered and non-covered MCF rubber as shown in Figure A6, the electrical sensitivity is reduced compared to the silicone oil rubber that covers the MCF rubber. Secondly, based on another experiment on coating using a liquid-rubber coating spray, the covering can be determined to be useless. In the case of the MCF rubber sensor produced using the procedure as shown in Figure A1 in the previous report [9], the MCF rubber sensor sprayed with liquid rubber-coating spray, which is an ordinary commercial liquid-rubber spray (LC-311SP, Jefcom, Co. Ltd., Osaka, Japan) exhibits the change that the 11.2 mV induced voltage at the production date becomes zero after 1 month. The MCF rubber sensor was consisted with the MCF rubber liquid with a hydrate as 1 g Ni, 0.75 g MF with 40 wt% Fe_3_O_4_ (W-40, Ichinen-Chemicals Co., Ltd., Shibaura, Japan), 3 g NR-latex (Ulacol), 3 g CR-latex (671A, Showa Denko Co. Ltd., Tokyo, Japan), 0.5 g TiO_2_ (Anatase type, Fujifilm Wako Pure Chemical Co., Ltd., Osaka, Japan), and 0.5 g hydrates Na_2_WO_4_·2H_2_O (Fujifilm Wako Pure Chemical Co., Ltd., Osaka, Japan), and MCF rubber without hydrate as 3 g Ni powder, 0.75 g MF (W-40), 3 g NR-latex (Ulacol), 3g CR-latex (671A), and 0.5 g TiO_2_. Irrespective of the material used to coat the MCF rubber or MCF rubber sensor, the instability of the aforementioned (b) is still present in the MCF rubber so that the construction in the MCF rubber sensor must be improved.

Incidentally, the present MCF rubber sensor is sensitive to both normal and shear forces, which has been presented in detail in our previous study [1,2]: the electric current passed between electrodes is changed alternatingly according to the deformation of the rubber by the application of the forces, whose passing phenomena is created by tunnel effect as shown in Figure A5a. For example, regarding the normal force, the electric resistance decreases abruptly by the application of minimal force as shown in Figure A6. This change is different by kinds of dopant involved in the MCF rubber as shown in the reference [2]. In addition, the electric resistivity of commercial pressure-sensitive electrically conductive rubbers (PSECRs) made of NR-latex (NR-latex), CR rubber (CR rubber) and silicon oil rubber has been presented, comparing with that of MCF rubber [1].

### 2.2. Combination of NR and CR, and Q

To resolve the instability highlighted in the previous section, the resistance of the non-diene rubber to water, heat, etc. must be considered. Moreover, we attempt to combine the non-diene rubber to diene rubber. In the present report, we deal with Q as a non-diene rubber. It is structured as the basis of dimethylpolysiloxane (PDMS) as shown in Equation A8 in the Appendix A, which is similar to oil. On the contrary, NR and CR are similar to water. Therefore, when PDMS and NR or CR are combined, polyvinyl alcohol (PVA) is used, which is anionic as shown in Equation (4). PVA is an emulsifier, then NR-latex or CR-latex and PDMS can be combined by emulsion polymerization as shown in Figure 1.

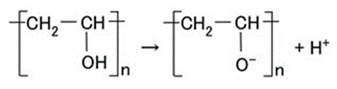
(4)

Q used in this report is KF96 (1 cSt, 50 cSt, 100 cSt, 1000 cSt) which is a silicone oil with the methyl group but not the rubber liquid, KE1400, and KE1300T that are hardened to be solid rubber using a curing agent. They are produced by Shin-Etsu Chemical Co. Ltd. (Tokyo, Japan). As for KE1400 and KE1300T, in general usage, they are solidified to be silicone rubber using a curing agent. KE1400 and KE1300T are improved because of solidification so that they are different from KF96; KE1400 and KE1300T are silicone oils with some silane, however, KF96 is a pure silicone oil without any silane. In the present study, we used only KE1400 and KE1300T without using the curing agent in the present combination. Therefore, the combination using KE1400 and KE1300T is different from that with KF96 from the perspective of the presence of silane. In the case of using PVA, we combined 3 g NR (Ulacol)- or CR (671A)- latex to the previously combined 3 g KF96, KE1400 or KE1300T with 3 g PVA. Figure 2 shows the appearance of the combination, which can achieve the same results by other combinations except for the figure. 

As seen in Figure 2a, there are many small granulated lumps that are non-combined ones of PDMS and NR- or CR-latex. By comparing Figure 2a,b, PVA can be seen to have a role in the combination between diene and non-diene rubbers. In the case of Figure 2d, after the combination of the KE1400 and CR-latex (671A) with PVA, MCF consisting of 3 g Ni and 0.75 g MF (W-40) was combined. The combination of diene and non-diene rubber latexes with MCF also results in a highly uniform dispersion.

Next, these combined liquids were electrolytically polymerized whereby a static magnetic field of 312 mT is applied to a pair of two stainless electrodes with a 1 mm gap using permanent magnets as paired opposites via the application of a constant electric field at 20 V, 2.7 A, and 5 min. The magnetic field strength can be determined by the production method such that has been presented in the previous studies [1]. It can be measure with Gauss meter probe which is an ordinary instrument for measurement of magnetic field. From trying many experiments, 312 mT has been confirmed to be optimal magnetic field strength in case of our present production method with 1-mm electrodes gap using permanent magnets as paired opposites via the application of a constant electric field at 2.7 A during 6–30 V, and 5–30 min. Therefore, we used 312 mT during the present study. The mass of each component is the same as that of Figure 2. In the case of using TiO_2_, a liquid containing 0.5 g TiO_2_ is combined beforehand to MF, combined with Ni, and then this liquid is combined to the combination of non-diene and diene rubber latexes with PVA. The combination of NR-latex and CR-latex is because the resistance to heat, ozone, etc. of CR-latex is superior to that of NR-latex. If we use just CR-latex as shown in Figure 3e,g,i, the electrolytically polymerized MCF rubber has such a thin thickness that it is difficult to handle at the subsequent procedure of producing it as a sensor style, as demonstrated in another experiment.

Figure 3 shows the surface of the MCF rubber on the cathode-side electrode after electrolytic polymerization. All surfaces are porous. The cause of creation of the porous by electrolytic polymerization in case of using PVA is due to the creation of hydrogen of water involved in the PVA, NR-latex and CR-latex. The electrolytic polymerization is correspondent to the electric degradation of water and the hydrogen is created on the cathode. In contrast, the surface of the anode-side electrode is non-porous because the vulcanization of the MCF rubber by electrolytic polymerization occurs on the anode. Using KE1300T or CR-latex, the porosity increases. 

To investigate the microscopic mechanism of electrolytic polymerization of the MCF rubber by combining diene and non-diene rubber latexes, we will consider the changes in temperature, voltage, and electric current of the MCF rubber during electrolytic polymerization as shown in Figure A7 in the Appendix A. In the case of using just NR-latex or CR-latex, the temperature due to electrolytic polymerization increases, the voltage decreases to a constant value, and electric current increases to be constant. However, for the combination of diene and non-diene rubber latexes, the temperature increase due to electrolytic polymerization is smaller compared to the case of only the NR-latex or CR-latex. The voltage is maintained at a constant at the applied voltage; the electric current increases temporarily and then decreases to zero. This result indicates that the electric current cannot easily flow inside the MCF rubber liquid because of the mixing with PDMS and PVA.

Regarding the instability of (1) as mentioned in the previous section, it was investigated using the experimental data on the mechanical property under tension as shown in Figure 4 using a commercial, compact tensile testing machine (SL-6002, IMADA-SS Co. Ltd., Toyohashi, Japan). All test specimens with rectangular parallelepiped shapes were 1 mm thick 10 mm wide and 10 mm long in the initial stage before tension. The maximum tensile force was 0.5 N and the compression speed was 100 mm/min. The data is for the second tension when this parameter was repeated many times to eliminate the Mullins effect. The electrolytic polymerization conditions are the same as those of Figure 3. The constituent of each MCF rubber is as follows: for KE1300T, 3 g KE1300T + 3 g NR-latex (Ulacol) + 3 g CR-latex (671A) + 3 g PVA + 0.75 g MF (W-40) + 3 g Ni; for KF96, 3 g KF96 (1 cSt) + 3 g NR-latex (Ulacol) + 3 g CR-latex (671A) + 3 g PVA + 0.75 g MF (W-40) + 3 g Ni; for Non-PDMS, 3 g NR-latex (Ulacol) + 3 g CR-latex (671A) + 0.75 g MF (W-40) + 3 g Ni. In the figure, the arrows indicate the cyclic tension of increasing and decreasing stress-strain. “0 day” indicates that the measurement was conducted on the same day as the production of the MCF rubber, and “after 2 days” indicates that the measurement was performed after being left in the air for 2 days from the production of the MCF rubber. Table 1 shows the mass of each MCF rubber sheet and the ratio of water evaporation (RWE) from the MCF rubber, which can be obtained from the reduction of the mass.

KF96 (1 cSt) has kinematic viscosity at 1 cSt and KE1300T at 7810 cSt because the molecular weight of PDMS of KF96 (1 cSt) is smaller than that of KE1300T. The PDMS has the ability to contain the remaining water in the MCF rubber. From Table 1, the RWE of KF96 (1 cSt) is larger than that of KE1300T. This is because the ability to contain the remaining water in the rubber deteriorates due to the smaller molecular weight of PDMS. In addition, KF96 (1 cSt) has more water in the rubber than KE1300T and the RWE of KF96 (1 cSt) is larger than that of KF96. Therefore, KF96 (1 cSt) easily hardens such that KE1300T is softer than KF96 (1 cSt) as shown in Figure 4. As a result, when PDMS is used, a higher molecular weight is required for the production of MCF rubber sensors. However, the RWE of KE1300T is slightly smaller than that of the non-PDMS as shown in Table 1. This is because PDMS has the ability to contain the remaining water in the rubber. Due to the water, KE1300T is softer than the non-PDMS as shown in Figure 4. As a result of this softness, a temporary decreasing change is represented by “A” in Figure 4. It depends on the deformation of the connection between PVA and the NR- or the CR-latex, and between the PVA and PDMS. With respect to the instability of (1), an argument will be presented after the following sensor production section.

### 2.3. Production of Sensor

Given that we evaluated the effectiveness of utilizing PDMS in combination with NR- or CR- latex and Q, we attempt to produce a sensor using PDMS. It is evident from Figure 3 that the MCF rubber that is electrolytically polymerized with PDMS is highly porous. Therefore, we infiltrated liquids using a vacuum as shown in (g) in Figure 5 and Figure 6

Prior to the infiltration, we prepared the electrolytically polymerized MCF rubber with PDMS as shown in (f), which is the same as that shown in Figure 3. The order of (a) and (b) in Figure 5 is significant. As for (b) in Figure 5, TiO_2_ is appropriate for combining with MCF before mixing Ni. The electrolytic polymerization of (e) in Figure 5 occurs for an applied electric field at 20 V, 2.7 A, and 5 min. The filtrated liquid can be anything, therefore, the MCF rubber can be fabricated using any intelligent liquid including battery acid, electrolyte, etc. Therefore, it is possible to produce a novel intelligent material using any intelligent fluid. In the present report, we used glycerin (99.5%, Kanto Chemical Co., Inc., Tokyo, Japan) because of the possibility of resolving instability (1). During filtration, the MCF rubber (f) is light enough to float upwards. Therefore, it must be held down by a weight in the infiltrating liquid.

In the case of instability (2), to abate the vulcanization of the MCF rubber from the time of electrolytic polymerization, which is not during the process but past the endpoint, the final procedure of drying the MCF rubber sensor shown in Figure 5 is important. The vulcanization of the MCF rubber achieved via both electrolytic and thermal polymerization. The latter is for drying.

However, the MCF rubber liquid with hydrate must be prepared as shown in (d) in Figure 5. The metal complex hydrate Na_2_WO_4_·2H_2_O and Na_2_MoO_4_·2H_2_O are suitable for the MCF rubber liquid combined with PDMS. Using the same MCF rubber sensor production process as in the previous study [8], MCF rubber liquid with hydrate (d) inserted between the sandwiched MCF rubber (g) together with thin electrical wires is electrolytically polymerized as shown in (h) and (i) in Figure 5. The electrolytic polymerization of (h) occurs at 30 V, 2.7 A, 5 min., and (i) 30 V, 2.7 A, 15 min in an air atmosphere. The constituent includes 3 g PDMS, 3 g PVA, 3 g NR-latex (Ulacol), 3 g CR-latex (671A), 0.5 g Na_2_WO_4_·2H_2_O, 0.75 g MF (W-40), and 3 g Ni for (c) in Figure 5, 1 g Ni for (d) in Figure 5. At all electrolytic polymerizations (e), (h), (i), a static magnetic field intensity of 312 mT produced by permanent magnets as paired opposites is applied to a pair of two stainless electrodes with a 1 mm gap.

Incidentally, the hard adhesion of the MCF rubber with hydrate to metal could be confirmed. Most metals adhere including gold, silver, and platinum, except for Cr and Ti. In the case when the glass coated with TiO_2_ is specific: such as when using KF96 and Na_2_WO_4_·2H_2_O, or KE1300T and Na_2_MoO_4_·2H_2_O, adhesion occurs. In the case when KF96 and Na_2_MoO_4_·2H_2_O, or KE1300T and Na_2_WO_4_·2H_2_O is used, there is no adhesion. The adhesion phenomenon is shown in Figure 7 for an anode electrode in the case of electrolytic polymerization. 

This is expected to induce the progression of metal coating to prevent corrosion and adhesive glue between the rubber and a metal as a novel adhesion technique. The present adhesion technique is superior to that proposed in the previous study [8], which is applicable to stainless steel, iron, lead, gold, silver, platinum, and glass coated with TiO_2_. Therefore, as shown Figure 8, the thin electrical wires with an outer diameter of approximately φ 1.3 mm with seven thin silver-gilt electrical wires with a diameter of approximately φ 0.1 mm and a length of approximately 5 mm adhere in the MCF rubber. As presented in a previous study [8], although the number of thin silver-gilt wires is a few and their diameter is very small, the wires cannot be detached from the electrolytically polymerized MCF rubber affixed around each thin wire.

Moreover, we can produce many fabricated layers by repeating a single layer between the electrolytically polymerized rubbers as shown in (g) in Figure 5. In the present report, we produced double layers via the production procedure shown in Figure 9. The internal structure is shown in Figure 10. For the electrolytically polymerized MCF rubbers such as (f) and (g) in Figure 5, it does not matter whether they are infiltrated or not.

### 2.4. Stability of MCF Rubber Sensor

Now, we investigate the effectiveness of the combination of PDMS, PVA, and filtration on the production of an MCF rubber sensor using the conditions as shown in Table 2. The aging change of the induced voltage of the MCF rubber sensor is shown in Figure 11. In the case in which sulfur is used, the MCF rubber is repeated regularly to be deformed by compression and has a high sensitivity to the application of pressure, as shown in Figure A8 in the Appendix A. This leads to the improved stability of the sensor under the application of a force. From Figure 11, by utilizing a combination of PDMS and PVA, it is possible to prevent the aging of the induced voltage of the MCF rubber sensor. Moreover, using filtration, it is possible to achieve a stable MCF rubber sensor. By using sulfur, the sensitivity can be enhanced although the MCF rubber becomes harder.

## 3. Sensitivity in Water

For a robot with the installed sensor working in a nuclear reactor building, we investigate the sensitivity of the MCF rubber sensor when immersed in water. The resistance to irradiation is very important, especially regarding the rubber degradation problem. The MCF rubber was determined to have a resistance to γ-irradiation in the previous report [9]. In addition, the sensitivity in water is significant because of the MCF rubber sensor’s resistance to water based on the observation that water does not evaporate easily from the MCF rubber, as shown in Table 1. 

The water in a reactor pressure vessel (RPV) is hot, therefore we must investigate the sensitivity of the sensor in hot as well as cold water. In addition, as observed in the Fukushima nuclear power plant accident in Japan in 2001, there are many kinds of debris such as metal, concrete, etc., in the saltwater of the RPV by the sea. Therefore, we must also investigate the sensitivity of the sensor for many kinds of body in saltwater. It is also necessary to consider both hard and soft materials, in addition to water with and without salt.

Firstly, we immersed a single MCF rubber sheet produced as shown in (f) in Figure 5 in water and the images are shown in Figure 12. In Figure 12a,b, the MCF rubber is solidified by drying in air. As shown in Figure 12c, the MF and water involved in the inner MCF rubber can be exuded because the electrolytic polymerization is proceeding towards instability (2), as mentioned in the previous section. The brown-colored region in the photograph represents exuded MF with water. However, by compounding CR-latex or TiO_2_, the MCF rubber cannot be easily exuded. As shown in Figure 12f,g, by combining PDMS and PVA, the rubber cannot be easily exuded and the weight trends to be low.

Next, we immersed the MCF rubber sensor using the adhesion technique as shown in Figure 5h,i, and this process is shown in Figure 13. Figure 13a represents the traditional electrolytic polymerization based on a previously reported adhesion technique [8], whereas MF and the water involved in the inner MCF rubber can be easily exuded. This is contrary to the single MCF rubber sheet as shown in Figure 12e. Even though TiO_2_ and infiltrating are utilized, the rubber cannot be easily exuded. As a result, we can use the MCF rubber sensor produced via a combination of PDMS and PVA in water. Its stability in water will then be investigated.

Next, we investigate the sensitivity of the MCF rubber sensor for a hard body made of acrylic resin and a soft body made of silicone oil rubber in water. The experimental procedure involved settling the MCF rubber sensors with single or double layers produced via the combination of PDMS and PVA on a half-round tube, as shown in Figure 14. Figure 15 represents the real appearance of Figure 14. The soft body is silicone oil rubber and the softness was adjusted by dilution with a thinner using the weight ratio KE1300T:thinner = 1:5. As shown in Figure A9 in the Appendix A, given that the MCF rubber sensor becomes softer as the temperature is increased, it becomes more compressible when in contact with a hard body. However, in case of contact with a soft body, the compressive displacement of the MCF rubber sensor does not increase because the soft body is also compressible.

The cause of settling in the half-round tube made of acrylic resin such that the MCF rubber sensor is bent as shown in Figure 14, is that the MCF rubber becomes highly sensible due to bending, as being confirmed by a separate experiment. Using the same commercial, compact tensile testing machine as in Figure 4, the MCF rubber sensor is repeatedly in contact with the body with a force up to 2 N and a compression speed of 10 mm/min. The constituent of our used MCF rubber sensor, as well as the electrolytic polymerization conditions, are the same as those of “a” as shown in Table 2 for both single- and double-layer types. Our used MCF rubber sensor is the one more than 30 days past from the production date because the effect of secular change of the MCF rubber sensor was excluded.

At first, we investigate the sensitivity of the MCF rubber sensor in the case of contact with the surface of water as shown in Figure 16. The water in the figure is hot, at a temperature of 50 °C. For the case of 25 °C, it was determined that the same results were obtained as for 50 °C based on a separate experiment. MCF rubber sensors with a single layer produced by a combination of PDMS and PVA were used. As shown in Figure 14, the MCF rubber sensor is in contact with the surface of the water. At the instantaneous moment when contact is made, water is absorbed by the surface of the MCF rubber sensor due to surface tension, as shown in Figure 17a. Therefore, the induced voltage of the MCF rubber sensor temporarily increases as shown by “A” in Figure 16. As such, the MCF rubber sensor is suitable as a contact sensor for water. When the entire surface of the MCF rubber is immersed in water as shown in Figure 17b, the induced voltage changes as “B” and “C” in Figure 16.

Figure 18 shows the results for the induced voltage of the MCF rubber sensor with a single layer produced by a combination of PDMS and PVA in water at 4 °C, 50 °C, and 75 °C respectively where 4 °C represents cold water. As seen from the figure, even if there is a long period from the production date, stability of the induced voltage to the pressure is observed, which is repeated regularly with respect to the pressure. As seen in Figure A9 in Appendix A, for the case of contact with a hard body, given that the MCF rubber sensor is more compressible as the temperature of the water increases, the induced voltage changes more significantly with compression. In the case of contact with a soft body, the induced voltage change should become smaller with compression because the MCF rubber sensor becomes less compressible. Nevertheless, the induced voltage changes with compression despite the higher temperature of the water. Therefore, it is evident from this result that the MCF rubber sensor has sensitivity when in contact with a soft body in hot water. In conclusion, the MCF rubber sensor is suitable as an underwater sensor in hot or cold water.

Figure 19 shows the results for the induced voltage from the MCF rubber sensor by comparing sensors with single and double layers produced by a combination of PDMS and PVA. The case of “50 deg” represents immersion in hot water at 50 °C whereas the case of “air” represents non-immersion in water, namely, in air at 25 °C. As seen from the figure, for the case of double-layer as well as the case with a single layer, even though a long period from the production date has elapsed, the stability of the induced voltage that is regularly repeated with respect to the pressure, can also be obtained regardless of the softness of the body in contact. The change of the induced voltage in the case of double layers is larger due to a higher sensitivity compared to the case of a single layer. In conclusion, the case with a double layer is also suitable for application as an underwater sensor.

Incidentally, the present MCF rubber sensor was confirmed to have temperature durability up to about 120 °C. This temperature is the maximum one of the MCF rubber during the electrolytic polymerization and changes by the production conditions of the MCF rubber involving its constituents, electrodes gap, applied voltage and electric current, magnetic field strength, etc. From many experiments, the MCF rubber was confirmed to have temperature durability up to about 120 °C.

Finally, regarding the case of immersion of an MCF rubber sensor with single layers produced by a combination of PDMS and PVA in saltwater, Figure 20 shows the results for the induced voltage produced by the MCF rubber sensor compared to the case of water without salt. The saltwater consisted of 5.7 wt% of salt. An experimental situation is assumed in the case of robots sensing in seawater in the RPV. In the case of salt water, osmotic pressure is acted on the surface of the MCF rubber sensor. However, the MCF rubber sensor has sensitivity to saltwater. Therefore, the MCF rubber sensor is applicable to the sensing of the retrieval of debris, etc. in Fukushima first nuclear power plant.

By the combination of PDMS and NR-latex, or diene type and non-diene type rubbers, which is a dominant key point in the present report, the durability for ambience of dehydration and waterproof can be brought about, and then it enables the typical applications under these ambiences involving water sensor as shown in the present report and rubber sensor used in a nuclear reactor building as shown in the previous report [9]. In addition, rubber sensor is also significant for valve sensor because of its elasticity for the normal and shear forces applied on the rubber which corresponds to pressure sensor and the role of sealing, for example, in the applications of fuel injection [25], gas valve [26], monitoring in a pipe [27], etc. As the present MCF rubber sensor has high sensitivity for the pressure and durability for water as shown in the above results, it is expectable for valve sensor. Especially, in case of the application to nuclear reactor facilities, we can surely look forward to many exciting utilization and development, for example, valve sensor, pressure sensor, velocity sensor, sealing, etc., because it is applicable in a water and resistible to degradation by irradiation.

## 4. Conclusions

To stabilize MCF rubber sensors, we propose the method of combining PDMS to NR-latex or CR-latex with PVA and filtration. Predominant parameters or conditions are the use of PVA and a voltage of 20 V and 30 V on the electrolytic polymerization of the rubber involving PVA as for production of the present rubber sensor. These optimal parameters have the rubber durable to various ambiences of dehydration, waterproof, etc.

The instability was examined both experimentally based on the results obtained during electrolytic polymerization, and theoretically based on tunnel theory. The stability was several requisites: (a) temporary stability during the sensing; (b) stability over a long span of time, which are related to other requisites: (1) non-evaporation of water from the MCF rubber; (2) abatement of the vulcanization of the MCF rubber from the time of electrolytic polymerization, which is not achieved during the process but when the endpoint exceeded; (3) prevention of the effect of deformation of the MCF rubber by Mullins effect. The combination method is also another novel combination of diene and non-diene rubbers. Silicone oil or rubber with PDMS can be combined with NR-latex and CR-latex by emulsion polymerization because PVA has hydrophilic and hydrophobic groups. In addition, by electrolytic polymerization with the combination of PDMS and PVA, the MCF rubber is porous so that it can be infiltrated with any liquid. This eventuates in the feasibility of producing a novel intelligent rubber using any intelligent fluid. We also propose a novel adhesion method for rubber and any metals that is superior to the adhesion technique previously presented in our study. By assembling the infiltrated MCF rubber sheets and by performing electrolytic polymerization using the MCF rubber liquid with hydrate based on the adhesive technique as presented in the previous study as well as using sulfur in NR-latex and CR-latex and fabricating multilayers, a sensor with many stabilities can be produced such as (1)–(3), with resistance to water from the cold to hot temperature range. The MCF rubber sensor is applicable for usage as a sensor for operation in a nuclear reactor building as well as an underwater sensor, based on its resistance to water as well as its irradiation resistance, as shown in the previous report.

## Figures and Tables

**Figure 1 sensors-19-03901-f001:**
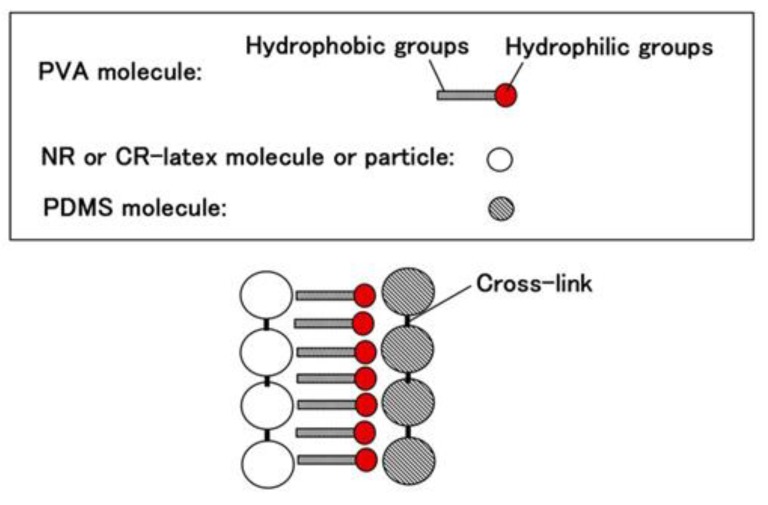
Schematic diagram of hydrophilic and hydrophobic group of PVA, and emulsion polymerization between PDMS and NR- or CR-latex.

**Figure 2 sensors-19-03901-f002:**
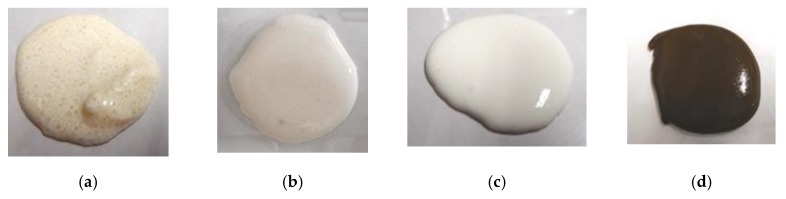
Images of liquid before electrolytic polymerization; (**a**) KF96 (1000 cSt) + NR-latex without PVA; (**b**) KF96 (1000 cSt) + NR-latex (Ulacol) with PVA; (**c**) KE1400 + CR-latex (671A) with PVA; (**d**) KE1400 + CR-latex (671A) + MCF with PVA.

**Figure 3 sensors-19-03901-f003:**
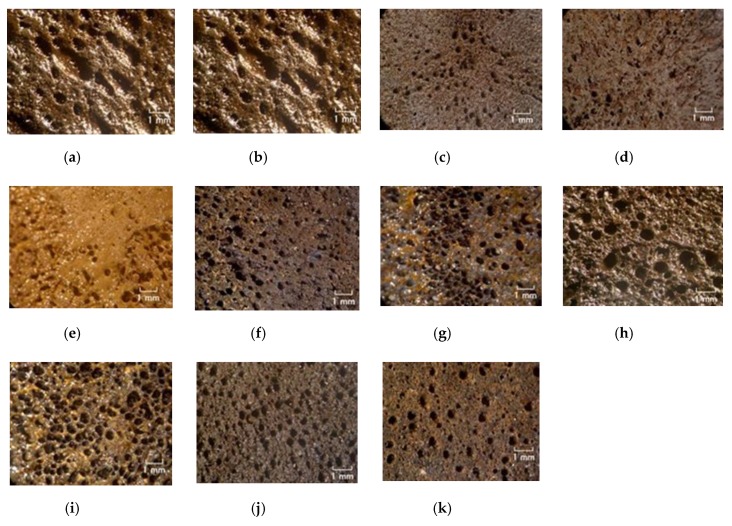
Photographs of surface on cathode-side electrode after electrolytic polymerization with PVA under a magnetic field; (**a**) KF96 (1000 cSt) + NR-latex (Ulacol); (**b**) KF96 (100 cSt) + NR-latex (Ulacol); (**c**) KF96 (50 cSt) + NR-latex (Ulacol); (**d**) KF96 (1 cSt) + NR-latex (Ulacol); (**e**) KF96 (1000 cSt) + CR-latex (671A); (**f**) KE1400 + NR-latex (Ulacol); (**g**) KE1400 + CR-latex(671A); (**h**) KE1300T + NR-latex (Ulacol); (**i**) KE1300T + CR-latex (671A); (**j**) KE1300T + NR-latex (Ulacol) + CR-latex (671A); (**k**) KE1300T + NR-latex (Ulacol) + CR-latex (671A) + TiO_2_.

**Figure 4 sensors-19-03901-f004:**
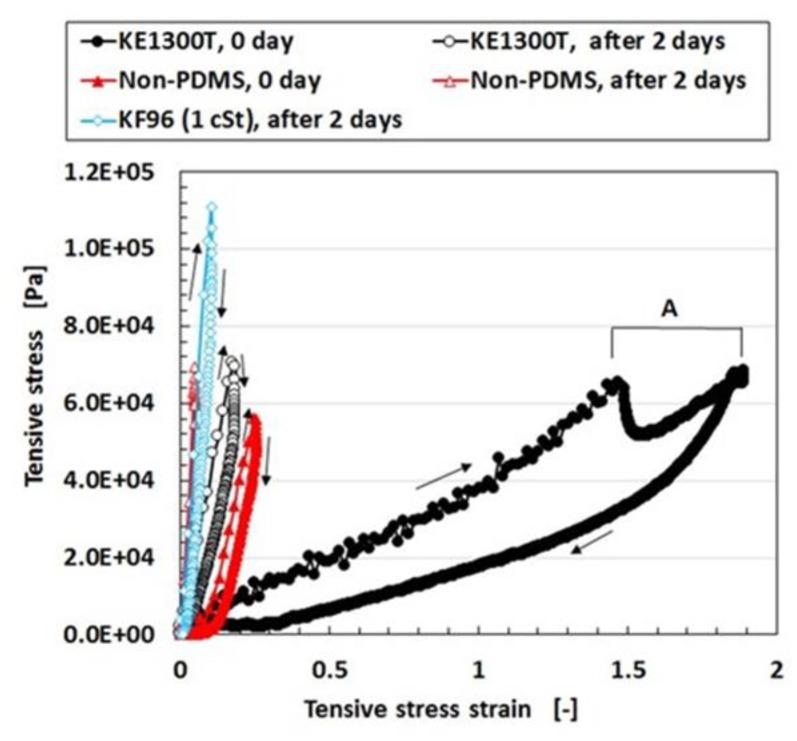
Relation between stress strain and stress in tension; as for KE1300T, 3 g KE1300T + 3 g PVA; as for Non-PDMS, without KE1300T or KF96 + 3 g, and PVA; as for KF96, 3 g KF96 + 3 g PVA; all has 3 g NR-latex (Ulacol) + 3 g CR-latex (671A) + 0.75 g MF(W-40) + 3 g Ni.

**Figure 5 sensors-19-03901-f005:**
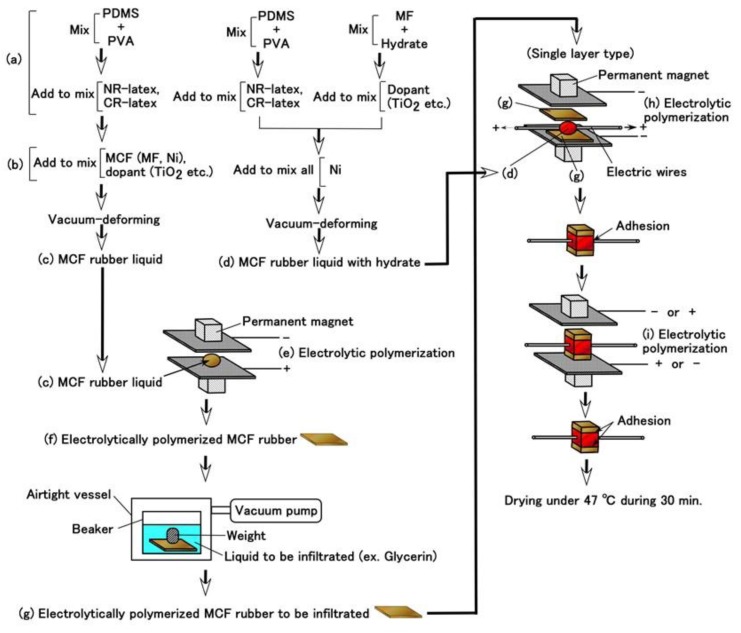
Procedure of the production of an MCF rubber sensor by combining PDMS and PVA.

**Figure 6 sensors-19-03901-f006:**
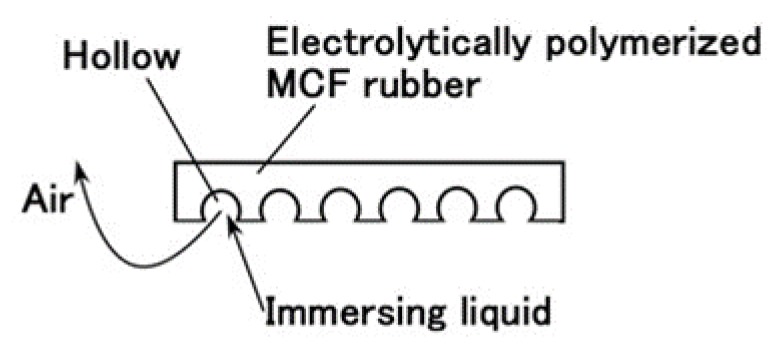
Physical model of infiltrated MCF rubber with a liquid.

**Figure 7 sensors-19-03901-f007:**
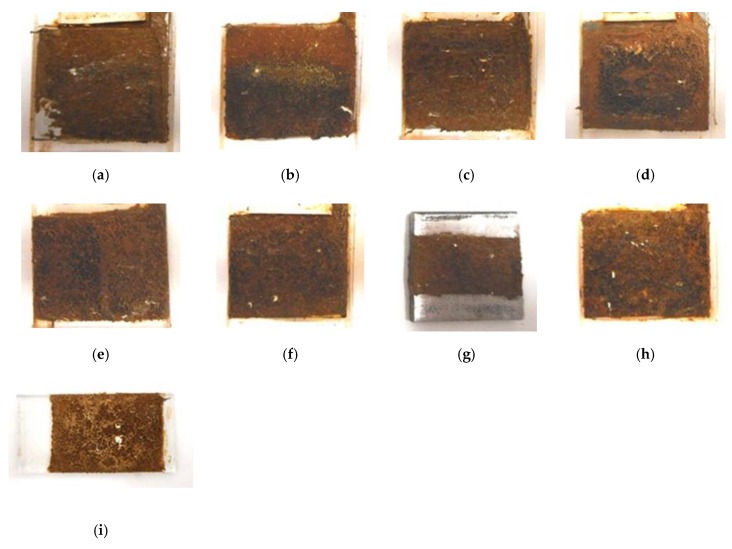
Images of surface of metal on cathode-side electrode after electrolytic polymerization with PVA, Na_2_WO_4_·2H_2_O, KF96 (1000 cSt) under an applied magnetic field; (**a**) aluminum; (**b**) stainless steel; (**c**) nickel; (**d**) zinc; (**e**) lead; (**f**) brass; (**g**) iron; (**h**) copper; (**i**) glass coated by TiO_2_.

**Figure 8 sensors-19-03901-f008:**
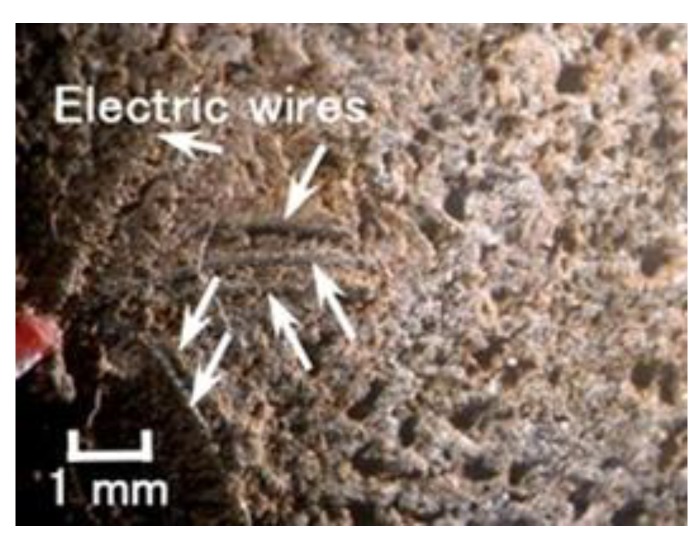
Image of electrical wires adhered to MCF rubber with hydrate inner MCF rubber sensor.

**Figure 9 sensors-19-03901-f009:**
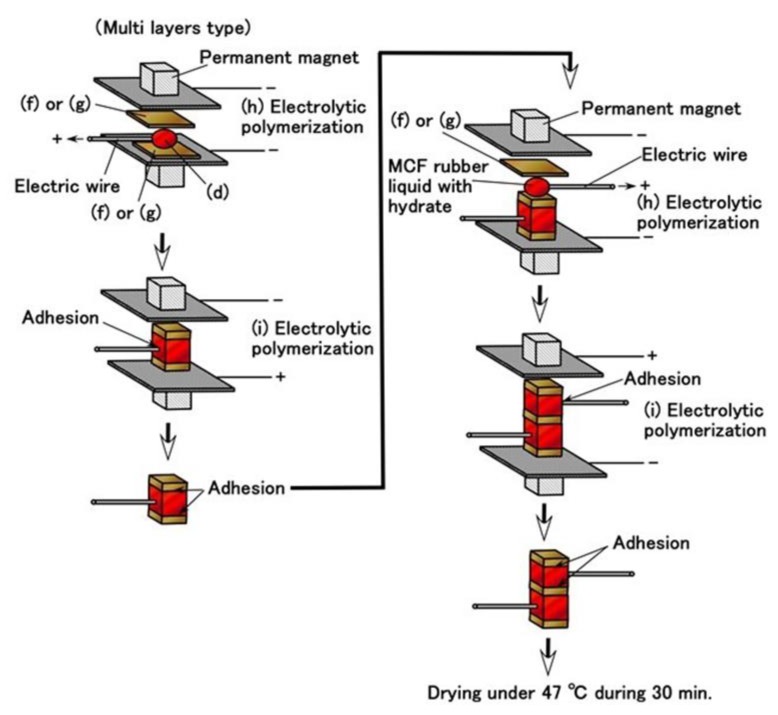
Procedure for the production of MCF rubber sensor with multi layers.

**Figure 10 sensors-19-03901-f010:**
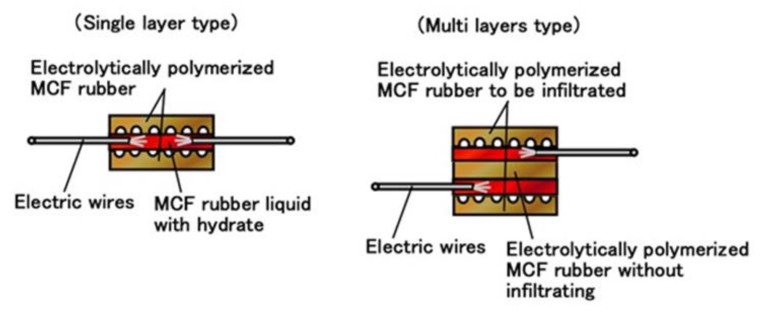
Schematic diagram of internal structure of MCF rubber sensor.

**Figure 11 sensors-19-03901-f011:**
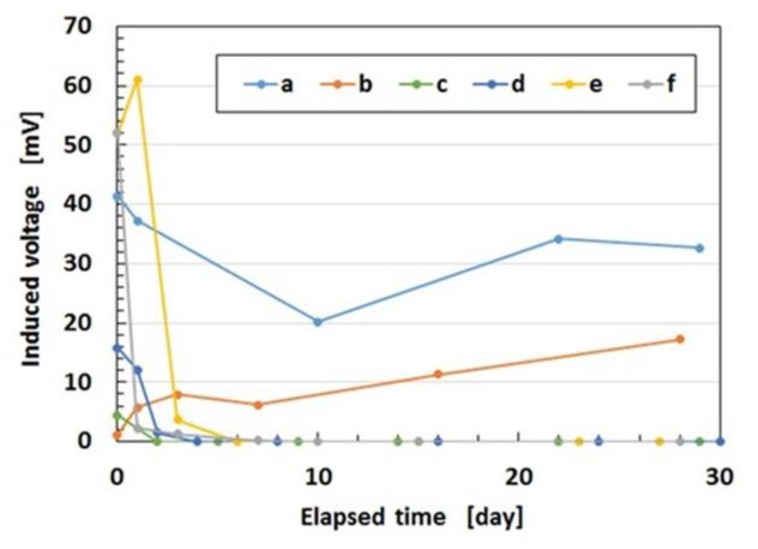
Change of the induced voltage with elapsed time.

**Figure 12 sensors-19-03901-f012:**
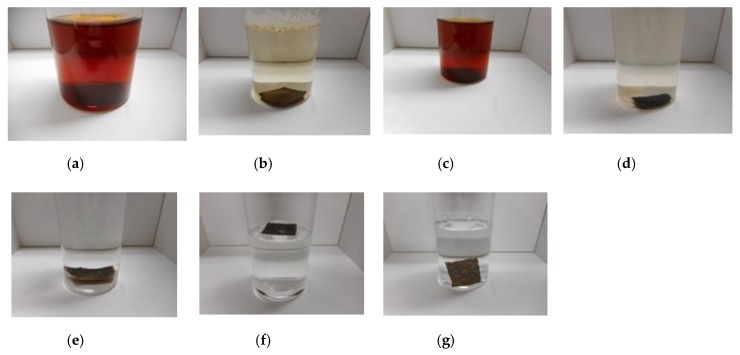
Images of single MCF rubber immersed in water; (**a**) NR-latex (Ulacol) by drying without a magnetic field; (**b**) NR-latex (Ulacol) by drying with a magnetic field; (**c**) NR-latex (Ulacol) by electrolytic polymerization with a magnetic field; (**d**) NR-latex (Ulacol) + CR-latex (671A) by electrolytic polymerization with a magnetic field; (**e**) NR-latex (Ulacol) + CR-latex (671A) + TiO_2_ by electrolytic polymerization with a magnetic field; (**f**) NR-latex (Ulacol) + CR-latex (671A) + KE1300T + PVA by electrolytic polymerization with a magnetic field; (**g**) NR-latex (Ulacol) + CR-latex (671A) + KE1300T + PVA + TiO_2_ by electrolytic polymerization with a magnetic field; all rubbers include MF (W-40) and Ni.

**Figure 13 sensors-19-03901-f013:**
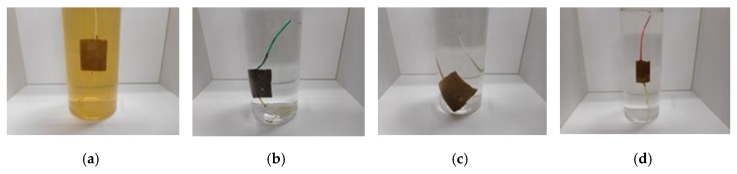
Images of the MCF rubber sensor immersed in water; (**a**) with TiO_2_ and without PDMS and PVA; (**b**) without TiO_2_, and with PDMS and PVA; (**c**) with TiO_2_, PDMS and PVA; (**d**) without TiO_2_, with PDMS and PVA, and infiltrated with glycerin; all rubbers include MF (W-40) + Ni + NR-latex (Ulacol) + CR-latex (671A) with an applied magnetic field.

**Figure 14 sensors-19-03901-f014:**
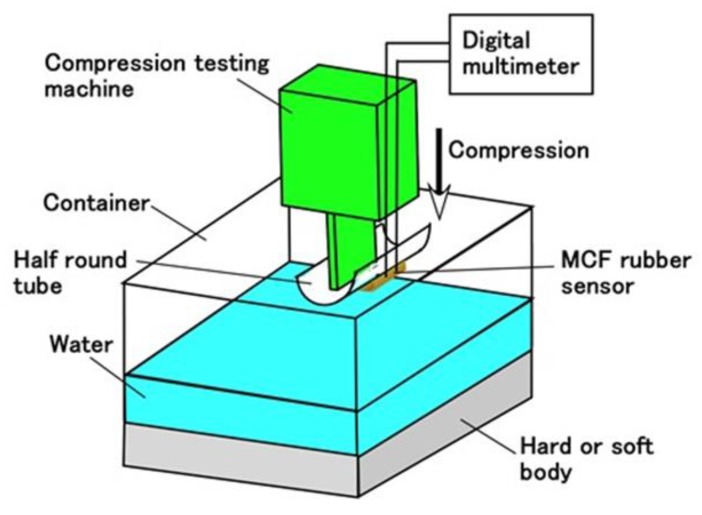
Schematic diagram of sensitive MCF rubber sensor in water under compression.

**Figure 15 sensors-19-03901-f015:**
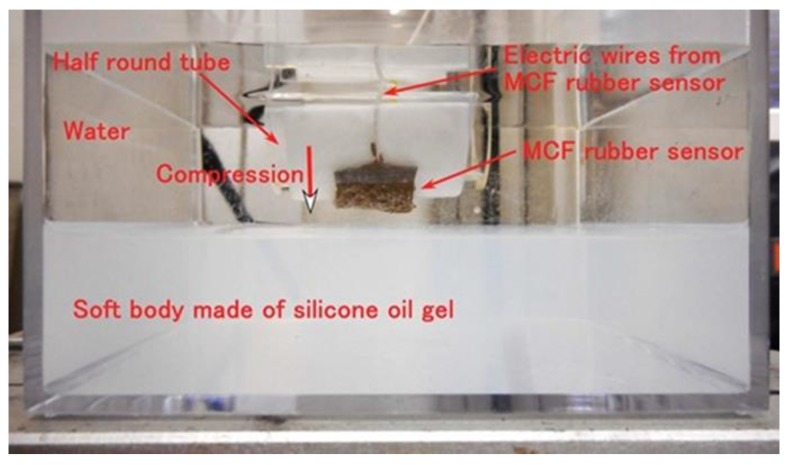
Corresponding image of the schematic shown in Figure 14.

**Figure 16 sensors-19-03901-f016:**
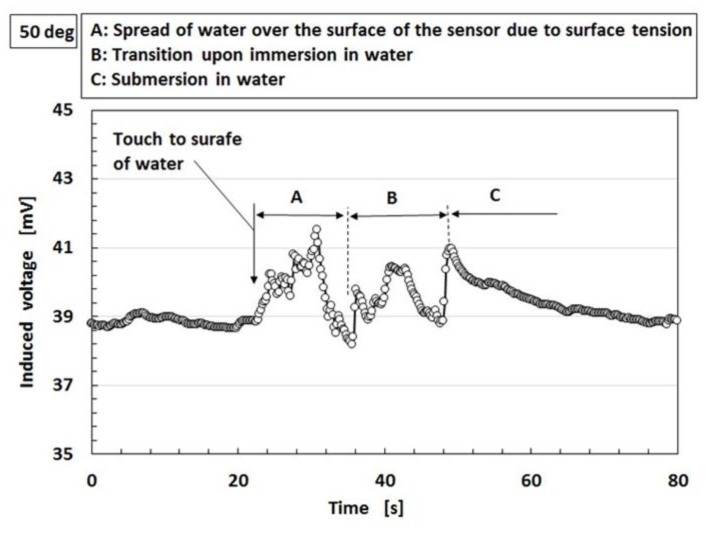
Change of induced voltage when in contact with water.

**Figure 17 sensors-19-03901-f017:**
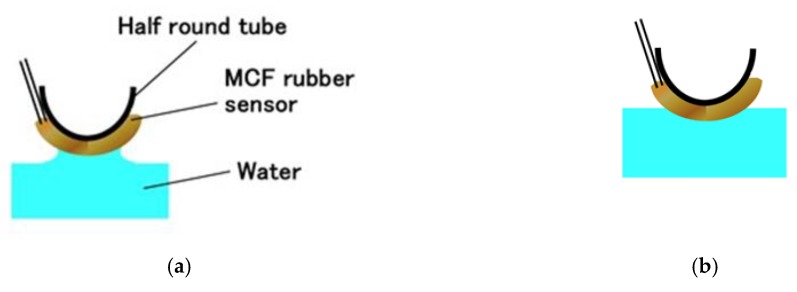
Schematic diagram for MCF rubber sensor in contact with water: (**a**) at time of sensor’s touching to water; (**b**) after sensor’s touching and then insertion to water.

**Figure 18 sensors-19-03901-f018:**
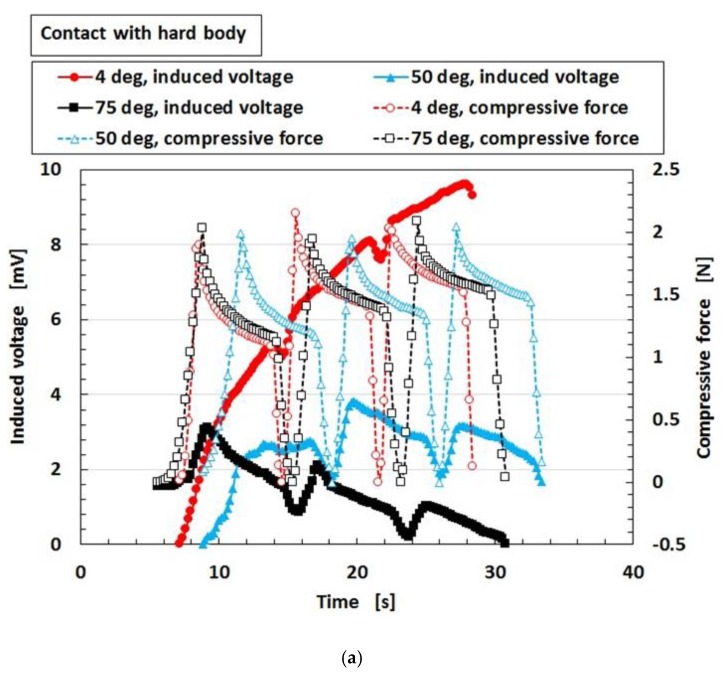
Induced voltage of MCF rubber sensor with single layer produced by combination of PDMS and PVA in water by compression: (**a**) to a hard body (**b**) to a soft body.

**Figure 19 sensors-19-03901-f019:**
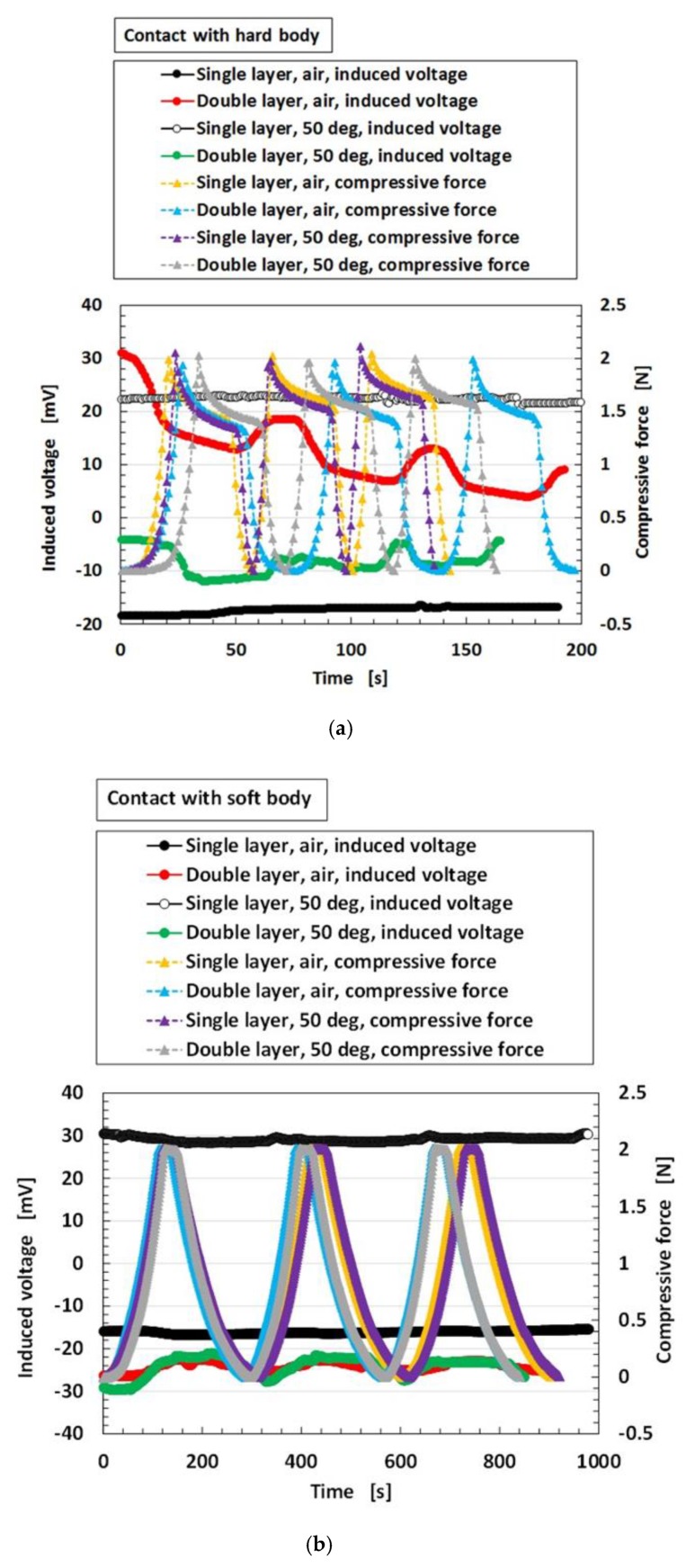
Comparison of induced voltage in water by compression between MCF rubber sensor with single layer produced by a combination of PDMS and PVA and that a double layer: (**a**) to a hard body (**b**) to a soft body.

**Figure 20 sensors-19-03901-f020:**
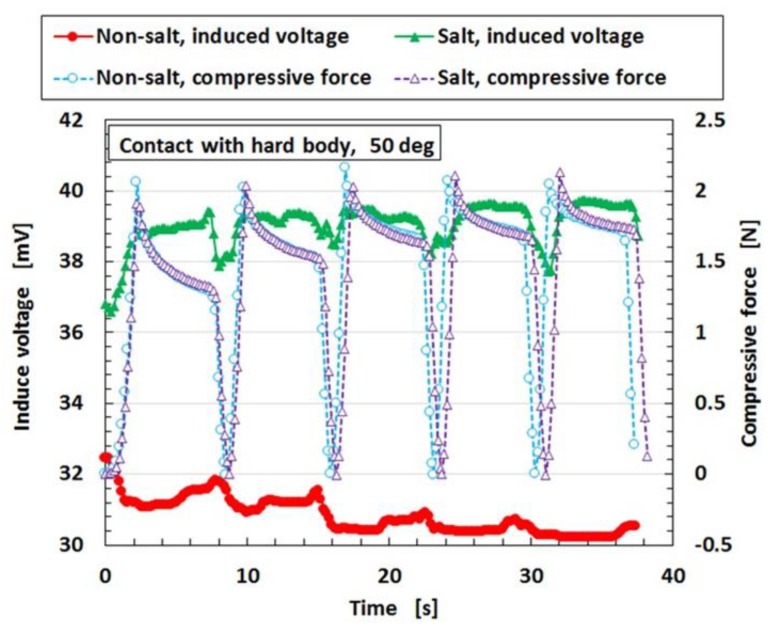
Induced voltage in salt water with compression.

**Table 1 sensors-19-03901-t001:** Change of the mass of MCF rubber sheet and ratio of water evaporation.

	Mass at 0 Day [g]	Mass After 2 Days [g]	Ratio of Water Evaporation (RWE) (%)
KE1300T	0.1573	0.1538	2.225
KF96	0.0678	0.0648	2.234
Non-PDMS	0.2372	0.2319	4.424

**Table 2 sensors-19-03901-t002:** Constitute of MCF rubber sensor in Figure 11 with 3 g NR-latex (Ulacol), 3 g CR-latex (671A), 0.75 g MF (W-40), and 3 g Ni for (f) or (g), 1 g Ni for (d) in Figure 9.

	Number of Layers	Filtration	Combination of PDMS and PVA	TiO_2_	Sulfur in NR-latex
a	2	yes	yes	yes	yes
b	1	yes	yes	yes	no
c	1	no	yes	yes	no
d	1	no	yes	no	no
e	1	no	no	yes	no
f	1	no	no	no	no

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
