# Peer review of "Development of a Magnetic Compound Fluid Rubber Stability Sensor and a Novel Production Technique via Combination of Natural, Chloroprene and Silicone Rubbers"

_sensors, 2019, doi:10.3390/s19183901_

Round 1

Reviewer 1 Report

In this manuscript, MCF rubber sensor was developed for usage as a sensor for operation in a nuclear reactor building as well as an underwater sensor, based on its resistance to water as well as its irradiation resistance. They proposed the combination method of PDMS to NR‐latex or CR‐latex with PVA and filtration to stabilize the MCF rubber sensor. The resulted MCF rubber sensor show excellent stabilities with resistance to water from the cold to hot temperature range. This manuscript reports good results and is also well written. I recommend this paper for publication in the journal Sensors after minor revisions.

Detailed comments include as follows:

How about the force-sensitivity of the MCF rubber sensor? Please give a comparison study with commercial pressure sensors.   There are so many keywords in the manuscript. It is better to delete some of them from the manuscript. What is the mechanism of the MCF rubber sensor using for operation in a nuclear reactor building?

Author Response

I appreciate for your valuable comments and suggestions for the present our report. According to their comments, we would like to reply for them as follows.

For comments :

/ The present MCF rubber sensor is sensitive to both normal and shear forces, which has been presented in detail in our previous study [1, 2]: the electric current passed between electrodes is changed alternatingly according to the deformation of the rubber by the application of the forces, whose passing phenomena is created by tunnel effect as shown in Fig. A5(a). For example, regarding the normal force, the electric resistance decreases abruptly by the application of minimal force as shown in Fig. A6. This change is different by kinds of dopant involved in the MCF rubber as shown in the reference [2]. In addition, the electric resistivity of commercial pressure-sensitive electrically conductive rubbers (PSECRs) made of NR-latex (NR-latex), CR rubber (CR rubber) and silicon oil rubber has been presented, comparing with that of MCF rubber [1]. These explanations are added in the text.

/ According to the suggestion, some key words are deleted.

/ In case of using the present MCF rubber sensor in a nuclear reactor building, it responds to the irradiation and dose not degraded by the irradiation, which are typical characteristics and different to ordinary rubber sensor. These are in detail presented in the 1st report that has been submitted simultaneously together with the present report to “Sensor” of MDPI. However, the first report was very regretfully rejected by peer-review, nevertheless the results are novel and suitable for science. Therefore, this suggestion on the using the present MCF rubber sensor in a nuclear reactor building is added in the text. 

Other revisions: As described above, the 1st report was regretfully rejected by peer-review in “Sensor” of MDPI, nevertheless the results are very novel and suitable for science. Therefore, the title involved “2nd report” has become meaningless. And then, the title is corrected and some explanations of conjunction between the 1st and 2nd reports in the text are corrected.

Reviewer 2 Report

Reviewer:

Sensors

Manuscript ID: sensors-571557

Title: Development of Magnetic Compound Fluid Rubber Sensor for Practical Usage: 2nd Report on the Stability and Novel Production Technique via Combination of NR, CR and Silicone Rubber  

Authors: Kunio Shimada*,Ryo Ikeda,Hiroshige Kikura, Hideharu Takahashi

Dear Editor:

The manuscript focused on the “Development of Magnetic Compound Fluid Rubber Sensor for Practical Usage: 2nd Report on the Stability and Novel Production Technique via Combination of NR, CR and Silicone Rubber”, which is new novel and very useful in sensor fields. It is recommended to accept after major revision. However, some parts need to revise, which are listed below as follows. The main points need to revise before publication.

[1] The new relate references are needed to add in the revised manuscript.

[2] The authors investigate many parameters in this study. What is optimal condition in this work? Please explain and add it in the revised manuscript.

[3] The grammar of English should be written more carefully in the manuscript; English must be checked and improved by Native English speaker.

[4] What are the important applications in this study? Please add in the revised manuscript.

[5] In Fig. 2, these combined liquids were electrolytically polymerized where by a static magnetic field of 312 mT is applied to a pair of two stainless electrodes with a 1 mm gap using permanent magnets as paired opposites via the application of a constant electric field at 20 V, 2.7 A, and 5 min. How to determine the strength of this magnetic field?

[6] Figure 3 shows the surface of the MCF rubber on the cathode‐side electrode after electrolytic polymerization. All surfaces are porous. Please explain why does the surface of the MCF rubber have porous?

[7] Does this sensor have the experimental result for high temperature durability?

[8] The sensitivity valve is very important to rubber sensor. Please provide the sensitivity value for the entire sensor.

Sincerely yours.

Author Response

I appreciate for your valuable comments and suggestions for the present our report. According to their comments, we would like to reply for them as follows.

For comments :

[1] All written relate references are latest. The published date in the references perhaps seems to be older. But the study on combination of PDMS and NR-latex, or diene type and non-diene type rubbers which is dominant topics in the present article, has not been conducted on for ordinary rubber.

[2] Plainly speaking, predominant parameters or conditions are the using of PVA and the voltage of 20 V and 30 V on the electrolytic polymerization of the rubber involving PVA as for production of the present rubber sensor. Because of many parameters in the electrolytic polymerization, this seems to be ambiguous. This explanation is added in the text.

[3] According to your suggestion, the grammar of English overall in the text is brush up by Native English speaker.

[4] By the combination of PDMS and NR-latex, or diene type and non-diene type rubbers, which is dominant key point in the present article, the durability for ambience of dehydration and waterproof can be brought about, and then it enables the typical applications under these ambiences involving water sensor as shown in the present report and rubber sensor used in a nuclear reactor building as shown in the previous report [9]. These explanations are added.

[5] The magnetic field strength can be determined by the production method such that has been presented in the previous studies [1]. It can be measured with Gauss-meter probe which is an ordinary instrument for measurement of magnetic field. From trying many experiments, 312 mT is optimal magnetic field strength in case of our present production method with 1-mm electrodes gap using permanent magnets as paired opposites via the application of a constant electric field at 2.7 A during 6 - 30 V, and 5 - 30 min. These explanations are added.

[6] The cause of creation of the porous by electrolytic polymerization in case of using PVA is due to the creation of hydrogen of water involved in the PVA, NR-latex and CR-latex. The electrolytic polymerization is correspondent to the electric degradation of water and the hydrogen is created on the cathode. These explanations are added.

[7] The present MCF rubber sensor was confirmed to have temperature durability up to about 120 ℃. This temperature is the maximum one of the MCF rubber during the electrolytic polymerization and changes by the production conditions of the MCF rubber involving its constituents, electrodes gap, applied voltage and electric current, magnetic field strength, etc.. From many experiments, the MCF rubber was confirmed to have temperature durability up to about 120 ℃. These explanations are added.

[8] According to your suggestion, the explanation that the sensitivity valve is significant for rubber sensor with some references is added.

Other revision: The 1st report that has been submitted simultaneously together with the present report to “Sensor” of MDPI was regretfully rejected by peer-review, nevertheless the results are very novel and suitable for science. Therefore, the title involved “2nd report” has become meaningless. And then, the title is corrected and some explanations of conjunction between the 1st and 2nd reports in the text are corrected.

Reviewer 3 Report

Comments on “Development of Magnetic Compound Fluid Rubber Sensor for Practical Usage: 2nd Report on the Stability and Novel Production Technique via Combination of NR, CR and Silicone” by Kunio Shimada et al.

In this work, the authors propose to combine the PDMS, NR-latex or CR-latex with PVA and filtration to stabilize the MCF rubber sensor. The stability of these sensors have been experimentally studied through electrolytic polymerization, and theoretically understood based on tunnel theory. The combination method also involves the diene and non-diene rubbers. In addition, by electrolytic polymerization with the combination of PDMS and PVA, the MCF rubber is porous so that it can be infiltrated with any liquid. In addition, the authors also propose a novel adhesion method for rubber and any metals that is superior to the adhesion technique developed by the same authors. The developed sensor has many stabilities such as 1) non-evaporation of water from the MCF rubber; 2) abatement of the vulcanization of the MCF rubber from the time of electrolytic polymerization, which is not achieved during the process but when the endpoint exceeded; 3) prevention of the effect of deformation of the MCF rubber by Mullins effect. Over all, the manuscript is well written and scientific sound. Therefore, the reviewer would like to suggest publication of this work.

Author Response

I appreciate for your valuable comments and suggestions for the present our report.

Round 2

Reviewer 2 Report

Dear Editor

According to the revised version, it can be accepted and published in this journal.

Sincerely yours.